# Schemes for Drug-Induced Treatment of Osteonecrosis of Jaws with Particular Emphasis on the Influence of Vitamin D on Therapeutic Effects

**DOI:** 10.3390/pharmaceutics13030354

**Published:** 2021-03-08

**Authors:** Filip Michalak, Sylwia Hnitecka, Marzena Dominiak, Kinga Grzech-Leśniak

**Affiliations:** 1Oral Surgery Department, Wroclaw Medical University, 50-367 Wroclaw, Poland; filip.michalak@umed.wroc.pl (F.M.); marzena.dominiak@umed.wroc.pl (M.D.); 2Maxillofacial Surgery Department, Wroclaw Medical University, 50-556 Wroclaw, Poland; sylwia.hnitecka@gmail.com; 3Department of Periodontics, School of Dentistry, Virginia Commonwealth University (VCU), Richmond, VA 23298, USA

**Keywords:** bisphosphonates, MRONJ, laser therapy, vitamin D, osteonecrosis

## Abstract

Drugs that inhibit bone resorption are prescribed most often by orthopedists, hematologists, or oncologists. Dental practice rarely draws attention to their importance and the effects they carry. The problem concerns mainly older people owing to oncological problems or postmenopausal consequences, but everyone can be at risk. Carefully conducted interviews and analysis of history and disease should always be performed before any action is taken by patients taking this type of medicine. Further action should consider possible complications and, above all, the risk of their occurrence. In this article, the most important issues related to the treatment of drug-induced osteonecrosis of the jaws (ONJ) are raised, including medication-related osteonecrosis of the jaw (MRONJ); conservative treatment, including the use of laser; and the impact of vitamin D supplementation on the overall treatment, prognosis, and prevention before complication, which is osteonecrosis of the jaw in the course of treatment with bisphosphonates and other drugs predisposing to MRONJ, such as denosumab and angiogenesis inhibitors. The degree of osteonecrosis is also critical, as it is possible to avoid surgical procedures for only conservative methods that sometimes bring good results. Surgical treatment of advanced stages is complicated and carries a high risk of error and complications. MRONJ is a disease that is easy to avoid, but it is difficult to treat and treatment sometimes leads only to a partial remission of the disease, not a complete cure.

## 1. Introduction

In daily dental practice, the necrosis of the jaws has been on the forefront for over a dozen years (MRONJ—medication-related osteonecrosis of the jaw) and is a relatively common complication in relation to bisphosphonate therapy, denosumab, and angiogenesis inhibitors. Osteonecrosis of the jaws associated with bisphosphonate therapy (BRONJ—bisphosphonate-related osteonecrosis of the jaw) was first described in 2003 [1,2,3]. However, it was found that not only this group of drugs, but also monoclonal antibodies against RANKL (activator for nuclear receptor factor κ B ligand) as well as inhibitors of angiogenesis, can increase the risk of MRONJ. The criteria for the diagnosis of drug-induced bone necrosis have been presented by the American Society of Oral and Maxillofacial Surgery (American Association of Oral and Maxillofacial Surgeons—AAOMS). The first of these is currently ongoing or past treatment with antiresorptive drugs and angiogenesis inhibitors; the second is the presence of visible, exposed, necrotic bone lasting longer than 8 weeks; and the last one is the lack of radiotherapy and the lack of tumor metastases in the head and neck area [4].

The incidence of MRONJ depends on many factors—both local and general. The main predisposing factors include all surgical procedures within the bone and jaws (about 60% of cases)—especially tooth extractions (of which 2/3 concern the molars in the mandible). The remaining 40% are not related to dental procedures [5,6]. Also important is poor oral hygiene accompanied by periodontitis, thin mucous membrane, injuries caused by unmatched prostheses [7,8], tooth decay, and co-existing abscesses [9]. General factors include the intake of corticosteroids, immunosuppressive drugs, and chemotherapy. Among chronic diseases, diabetes and rheumatoid arthritis (RA) are important. An increased risk of necrosis was also observed in patients after the sixth decade of life and active smokers [10]. The duration of therapy, the route of administration, and its type and dose [7] are also important in the development of ONJ (osteonecrosis of the jaws).

Despite the etiology and molecular pathogenesis of this disease not being fully explained, two theories describing the pattern of the disease are currently dominant. The first describes it as a primary bone infection with secondary dissemination to lower-lying soft tissues. The second (less preferred) talks about the primary infection of soft tissues with secondary bone involvement [11].

The clinical classification proposed by Ruggiero et al. is based on three stages: first stage—the asymptomatic unveiling of bones and soft tissues; second stage—exposed bone, infection, and soft tissue pain; and third stage—with exposed bone, pain, pathological fractures, and soft tissue infections [12]. Necrotic lesions are more common in the mandible and are located there in 65% of cases [13].

It was found that the best method of prevention is MRONJ failure to tooth extractions for their rescue conservative, or even root canal treatment, as the risk of ONJ in patients taking bisphosphonates for osteoporosis is 0.01% to 0.04%, and it increases in the case of tooth extraction from 0.09% to 0.34%. The situation is worsening in oncological patients. Here, the risk of disease is estimated at 0.88% to 1.15%, which, in combination with tooth extractions, gives chances of 6.67% to 9.1% of the disease occurring, which should undoubtedly persuade medical staff to use methods of prevention and make efforts to avoid this complication [14].

Currently, there is no clearly defined scheme for dealing with specific stages of MRONJ. Several studies are constantly being conducted that compare individual therapeutic methods and their effects. Treatment is difficult and requires proper planning of individual activities, taking into account a detailed interview with the patient and his current state of health and past sickness. In principle, treatment can be divided into non-surgical and surgical. Non-surgical treatment is mainly based on the improvement of oral hygiene, periodic dental check-ups, mouthwash with chlorhexidine, antibiotic therapy, and low-level laser therapy. The surgical treatment, in turn, involves the removal of necrotic lesions by conventional surgery and using erbium-doped YAG laser (Er:YAG) to evaporate necrotic bone [15]. Among the prevention and therapy methods, it is also worth mentioning vitamin D, which has a positive effect on the entire osteogenic system, and its invaluable properties are also used in patients with MRONJ. The positive effect of its supplementation during treatment and its deficiency in the vast majority of patients is widely emphasized. Deficiency of this vitamin is important and is a risk factor for the development of ONJ in patients treated with bisphosphonates (BPs), and its supplementation and maintaining a normal level can significantly affect the risk of disease [16].

## 2. Discussion

### 2.1. Drugs Predisposing to MRONJ

Bisphosphonates (BPs) as phosphoorganic chemical compounds have found a wide application in the therapy of disorders related to the skeletal system. Currently, they are used in therapies for diseases such as osteoporosis, multiple myeloma, Paget’s disease, bone metastases, or otosclerosis. Their action is based on inhibition of bone resorption and inhibition of osteoclast formation from the monocyte line. They can increase apoptosis of osteoclasts and reduce the activity of the osteoblasts. They show a strong affinity for calcium, and thus combine with bone hydroxyapatites. Their long half-life has a negative effect on any effects of bisphosphonate therapy, despite its cessation. They are able to survive hydroxyapatite-related up to 10 years. Osteoclast studies show that jawbones show increased BP uptake compared with other long bones [17]. Under natural conditions, the osteoclasts responsible for the resorption of dead bone cells and release of cytokines include BMP (bone morphogenetic protein) and ILG 1, 2 (insulin-like growth factors), which induce differentiation of mesenchymal cells into active osteoblasts [18]. Interruption of this homeostatic cycle causes excessive inhibition of resorption and accumulation of dead cells, resulting in necrosis of the jaw bone [19]. Not all bisphosphonates are equally effective in the development of osteonecrosis of the jaws. The nitrogen-containing bisphosphonates (alendronate, pamidronate, zoledronate, ibandronate, and risedronate) are particularly significant. They intensify inflammation and necrosis development through activation of mediators of inflammatory factors, among others, interleukin (IL)-1 and IL-6 [2,20]. Used in cancer treatment, they effectively inhibit cancer cells from bone involvement and induce their apoptosis. They also demonstrate antigenic properties by lowering the concentration of vascular endothelial growth factor (VEGF), which has a positive influence on the occurrence of osteonecrosis of the jaw [21]. The most populous group of patients taking bisphosphonates are people suffering from osteoporosis. However, they are in the group with a low risk of developing osteonecrosis of the jaw owing to the most common oral route of administration and a short cycle of treatment. The situation is similar when the patients are treated for Paget’s disease [7]. On the other hand, high-risk patients are oncological patients suffering from multiple myeloma, malignant hypercalcemia, and bone metastases in a compound of at least breast or prostate cancer. Drugs have been shown to increase the risk of developing ONJ by four times and can affect up to 90% of patients. Individual patients receiving oral drugs have been observed to have an occurrence of necrosis of the bone in 7.8% of cases, which appear at the earliest after 5 years from the start of treatment [3,22]. The route of administration is particularly important because of the absorption of the drug. Assuming, with oral bisphosphonates, only 1% of the dose is absorbed through the digestive tract; in the case of the intravenous route of administration, it is up to 50% bioavailable, and connects to the bone matrix, hence the divergence and the risk of individual patients [23]. Acil Y. et al. in one in vitro study demonstrated that intravenously administered zoledronate, pamidronate, and oral alendronate affect the reduction of cell proliferation and collagen production of the oral cavity, osteoblasts, as well as bone sarcoma cells (SaOS-2). Zoledronate has the hardest impact on the inhibition of these processes [24]. Among the side effects is also the occurrence of hypocalcemia; the strongest action is also by zoledronate, whose action can be increased by a lack of vitamin D [25,26,27].

Another drug predisposing to the occurrence of ONJ is the monoclonal antibody, an inhibitor of RANKL (denosumab). This is the anti-resorptive drug inhibiting function of osteoblasts and decreasing density of bone. It is used in patients with osteoarthritis and bone metastases [28]. In 2010, the necrosis of the jawbone associated with the intake of this drug was first described (DRONJ—denosumab related osteonecrosis of the jaw) [29]. DRONJ can occur at lower doses of the drug compared with BRONJ, the risk of which usually depends on the dose of the additive. The half-life of this drug is about 32 days. After the first dose, inhibition of osteoclast activity occurs already after 6 h and is maintained at 6 months after the last dose of a drug. Denosumab does not accumulate in the bone; hence, DRONJ is less intense than BRONJ and responds better to conservative treatment [30]. One analysis showed that the overall incidence of ONJ in cancer patients receiving denosumab was 1.7%. In addition, the increased risk with additional predisposing factors such as tooth extraction, poor oral hygiene, use of mobile phones, and chemotherapy [31] were highlighted in particular.

Anti-anginal drugs are the third group of drugs associated with an increased risk of osteonecrosis of the jaws in combination with antiresorptive therapy. Anti-angiogenesis inhibitors are increasingly being used to treat many malignancies, including ovarian cancer, metastatic renal cancer, breast cancer, colorectal cancer, lung cancer (NSCLC—non-small-cell lung carcinoma), and glioblastoma multiforme. Angiogenesis inhibitors can be divided into three main groups based on their mechanism of action: monoclonal anti-VEGF (e.g., Bevacizumab), bait VEGF receptors (VEGF decoy), and VEGF trap (e.g., aflibercept), and inhibit small molecule tyrosine kinase (TKI) (e.g., sunitinib, cabozantinib, and sorafenib). In addition, the target of mTOR (mammalian target of rapamycin) inhibitors also appears to have anti-angiogenic activity by inhibiting the production of VEGF and platelet-derived growth factors (PDGFs) [32].

### 2.2. Non-Surgical and Surgical Treatment

The MRONJ treatment regimen remains a contentious issue and is considered in many respects, taking into account clinical aspects of improvement and patient well-being. Although numerous results may suggest an advantage of surgical treatment in patients with MRONJ, the AAOMS position considers primary prophylaxis to be the most important factor of treatment and prevention. If, however, the disease occurs, surgical treatment is recommended only in stage II or III resistant to non-surgical treatment (Ruggiero et al., 2014).

In the subjects performed by G. Favia et al. [33], the results of surgical and non-surgical treatment were presented. Patients were divided into two groups: G1 (surgical treatment) and G2 (non-surgical-including diode laser for general health reasons). The results observed over 18 months showed complete cure of patients with stage I and II ONJ after implementing surgical and 86.5% of the total cure ego present in patients with stage III. The remaining 13.5% showed a reduction in the severity of osteonecrosis of the jaws from stage III to I (this concerned oncological patients who could not interrupt corticosteroid/antiresorptive therapy). In turn, patients from the G2 group did not show complete recovery. Only two patients with stage III and II changes, respectively, showed a lower stage transition, while a patient showed deterioration from stage II to III [33].

Surgical treatment of ONJ is often applied with autogenous growth factors. PRP (platelet-rich plasma) and PRF (platelet-rich fibrin) are biocompatible and have been successfully used to improve the healing of extraction sockets in patients receiving BP and those suffering from ONJ. They improve angiogenesis; therefore, the risk of osteonecrosis, where the main predisposing factor is the lack of vascularisation, is significantly reduced. Growth factors are responsible for stimulating undifferentiated stem cells at the wound site, stimulating their mitotic divisions, differentiating towards osteoblasts, and supporting healing processes [34].

An effective support for the treatment of defense is also hyperbaric therapy, especially in combination with surgery and antibiotic treatment, especially in the patient taking oral BP [35].

The specific advantages and disadvantages of surgical treatment are summarized in Table 1.

### 2.3. LLLT (Low-Level Laser Therapy)

Recently, in dentistry, diode lasers called LLLs (low-level lasers) have gained popularity. The term is used to describe soft, medium, low energy, and cold lasers. In the international definition, LLLT does not cause an increase in tissue temperature above 36.5 °C or above body temperature. The waveband has a length of 500–1200 nm [36,37]. It turns out that they can contribute to the improvement of treatment conditions and positively influence the remission of changes in the course of MRONJ.

Shin et al. examined the effect of LLL therapy on bisphosphonate-treated osteoblasts [38]. Human fetal osteoblastic cells were first subjected to alendronate, and then the effect of LLL irradiation was examined. Three osteoblast cytokines were analysed in detail: RANKL, OPG (osteoprotegerin gland), and M-CSF (macrophage-colony stimulating factor). It has been demonstrated that alendronate at concentrations above 50 μM decreases cell survival. The analysis shows that the solution of this condition may be the laser because the laser (Ga, Al-As) with a wavelength of 808 nm clearly increases cell survival, especially after 72 h. The positive effect of irradiation was particularly noted in the RANKL and M-CSF cytokines. A low-energy laser leaked the most to OPG expression. These results show that LLLT can be successfully used in the treatment of MRONJ [38]. A 65-year-old female patient described by Momesso et al. confirms that a relatively simple scheme, based on a combination of antibiotic therapy, LLLT, and rinses with a 0.12% solution of chlorhexidine (CHX), can lead to a complete cure. The implant in the posterior section of the jaw disintegrated; a 5-year history of intravenous administration of sodium alendronate was found 5 years ago. The patient was classified as stage II MRONJ. After the implant was removed, an 8 weeks treatment schedule was used, including LLLT three times per week, clindamycin 300 mg three times daily, and rinse with 0.12% chlorhexidine during this period. Already, after 6 weeks, a significant improvement was noticed and observations over 6 months showed complete healing of soft and hard tissues [39]. Certain treatment regimens and their use were evaluated by Vescovi et al. In most cases, the research shows that the use of LLLT alone or in combination with traditional surgery increases the percentage of total healings and health improvements. This is particularly evident in patients in stage II and III ONJ [38].

### 2.4. Nd:YAG Laser and Er:YAG

Vescovi et al. [38] examined the effect of Nd:YAG laser irradiation in a group of 28 patients affected by drug-induced bone necrosis. Fourteen patients received only laser biostimulation, without taking any antibiotics and without undergoing surgical procedures. Nine of the 14 people displayed a complete cure—meaning no signs of pain, infection, and exposed bone in the mouth, as if to confirm the positive effect of this therapy on the effects of treatment [38]. An important advantage of the Nd:YAG laser is not only antibacterial, but also antifungal activity, which has been documented based on the assessment of the irradiation effect on the colonies of *Candida albicans*-fungus, which is a commensal of the gastrointestinal tract, but in patients with impaired immunity, it can be difficult to treat and cause opportunistic infections. The effect of beam irradiation with different parameters was assessed: in the first group, 0.25 W, 10 Hz, 15 s, 3 J; and in the second group, 1 W, 10 Hz, 60 s, 59 J. Significant reduction of *C. albicans* immediately after irradiation was observed in both groups: in the first group, 20–54%, and in the second group, 10–60%, which undoubtedly indicates its effectiveness. Not without significance is the fact that the laser Nd:YAG also contributes to stimulating the healing process and reducing inflammation, including pain [39,40].

The results also demonstrate the significant effect of the Er:YAG laser. Statistically, a laser treatment using Er:YAG has the percentage of healing regardless of the stage of necrosis, as well as the patient’s history. This method is gaining popularity over conventional surgery. It is characterized by less thermal damage to tissue and faster healing. Laser Er:YAG does not cause coagulation or carbonation of tissues owing to the wavelength. A clean and precise cut is combined with bone ablation, thanks to which more cells are more likely to join. Er:YAG is not inferior to conventional surgery and can be used to remove large and deep fragments of necrosis with altered bone. It is possible that the combination of Er:YAG laser surgery with laser biostimulation Nd:YAG will soon become the gold standard in the majority of patients entering treatment at any stage of the disease, thus eliminating the percentage of treatments using sharp machine tools [41].

The effectiveness of the combination of Er:YAG and Nd:YAG lasers in the surgical treatment of bisphosphonate osteonecrosis encourages their use instead of conventional surgical instruments. One article describes the case of ONJ treated surgically in a three-step procedure: (1) Ablation of the inflamed bone with the Er:YAG laser (LightWalker^®^ Fotona, Dallas, TX, USA): 200 mJ/cm^2^, 2 W/cm^2^, 10 Hz; (2) Reduction of bacteria in the operating field using the Nd:YAG laser: 1.5 W/cm², 15 Hz, with the next application of a collagen sponge with gentamicin; (3) Photobiomodulation—externally—in treatment site laser light (Lasotronix^®^ Smart M, Piaseczno, Poland) 635 nm, 3 J/cm^2^ at the point of application, 25 s for 2 weeks (three times a week). Observations after 12 weeks proved the proper healing of soft tissues and the performed radiological examination showed no symptoms of decreasing the thickness of the bone plaque, which could indicate it will become a chronic condition after fire [42].

It is currently not a requirement to treat each patient with the Er:YAG laser. Simpler patterns are also acceptable and give a complete cure, but the advantages of using them undoubtedly speak to the appropriateness of implementing this method. Non-surgical therapy with the use of lasers is presented in Table 2, with particular attention to the advantages and disadvantages.

### 2.5. Antibiotic

A lot of attention is paid to antibiotic therapy, which in itself is not an ideal treatment for MRONJ, but it is a necessary complement. Bacteria show increased bone adhesion covered with bisphosphonates [43]. Research shows that bacterial flora in patients with ONJ is dominated by *Actinomyces*, *Eikenella*, and *Moraxella*. Therefore, neither amoxicillin nor clindamycin are the best first-line drugs. Preferably with a check, in this case, in phenoxymethylpenicillin at 500 mg 4× daily; in the case of allergy, 1× doxycycline at 100 mg per day; and patients who are resistant to the above treatment will be administered with metronidazole at 200 mg 3× daily [44]. Antibiotic therapy is mainly recommended for patients in stages II and III of MRONJ as support for surgical treatment and relief of local symptoms. Currently, there are no clear indications as to a specific antibiotic in relation to the disease MRONJ. M. Zirk and his associates [45] examined retrospective study bacterial species and yeasts found in necrotic bone specimen in 98 patients. He observed the bacterial flora changes during treatment (including surgery) of the patient. The dead bone contains both Gram + and Gram bacteria, with the predominance of the former in a ratio of approximately 60/40%. Apart from bacteria, every fourth sample of necrotic bones shows the presence of fungi, mainly from the Candida family [45].

According to the accepted standards, the first-line antibiotics in the case of MRONJ are penicillin antibiotics, including amoxicillin, with clavulanic acid at a dose of 875 mg/125 mg twice a day; however, the studies cited by M. Zirk show that, in 70% of the samples taken from patients, there was at least one species of bacteria resistant to penicillins, which further hinders effective treatment [45,46].

Considering these data, two schedules for the administration of penicillins emerge. In the first case, they are combined with B-Lactamase inhibitors, while alternatively, penicillins can be used in combination with metronidazole [47]. In the literature, besides clindamycin as a second-line antibiotic at a dose of 3 × 300 mg, there are also fluoroquinolones, i.e., moxifloxacin at a dose of 400 mg once a day during treatment [45]. Preparation for the procedure should include appropriately selected antibiotic therapy approximately 3 days before the procedure and continued minimum 7 days [48]. However, the dominant view in most studies is that treatment with antibiotics should be long-term, preferably until the healing process is completed [49,50,51]. It is most advantageous to perform an antibiotic for targeted antibiotic therapy and the best treatment effects, while broad-spectrum antibiotic therapy is used in cases where treatment should be initiated as soon as possible [52]. It should be added that antibiotic therapy is an important aspect not only of the treatment of the bone necrosis itself, but also of prophylaxis, e.g., during tooth extraction in patients at risk, and its proper use may prevent the development of necrotic changes in the jaw and mandible [45].

Procedures aimed at reducing bacteria in the course of ONJ are a very important element in the management of this disease, especially in patients with additional systemic diseases, who may develop a life-threatening complication of sepsis [53]. Systemic antibiotic therapy is implemented in patients with ONJ before surgical procedures. However, considering the bacterial flora in the tissues adjacent to the necrotic bone, it has been shown that orally administered antibiotics (tetracycline, doxycycline, ciprofloxacin, and amoxicillin) show limited efficacy. There are changes in the population of microorganisms, but not enough to reduce or eliminate the infection. One of the main factors is reduced vascularization of the bone tissue and necrosis as well as bacterial colonization in the form of biofilm [54,55,56]. The local action recommended at the early stages of the disease (stage I ONJ) is the use of oral antibacterial rinses, in turn in more advanced cases where surgical procedures are recommended (stage III ONJ); sometimes, they include the use of antibiotics locally.

Drug delivery scaffolds are a new trend that, according to research, shows significant positive effects, both local and systemic, compared with conventional parenteral or oral antibiotic therapy. Thanks to the creation of a 3D scaffold with special macro and micropore characteristics, it is possible to have an osteoconductive effect and release drugs by diffusion. In the case of biodegradable forms, the very process of biodegradation and its speed contribute to the local delivery of a specific amount of the drug. A difficulty in the treatment of MRONJ is poor vascularization of the bone tissue, which results in poor penetration of the antibiotic administered systemically in high doses and the development of bacterial resistance [57]. Drug-eluting matrices have a number of benefits. They enable the local delivery of appropriate amounts of the drug at a high concentration in a specific time without overloading the places where it is unnecessary and, thanks to their structure, they promote tissue regeneration. The drug itself can be administered in microcapsules or plated on a polymer scaffold or 3D porous matrix. Another advantage is the ability to deliver more than one active ingredient topically [58]. An example is collagen guts with gentamicin [35,42]. Jachewicz et al. [35] argue their use for two reasons. First, the experience of the authors shows that the main problem during the healing period is the exposing of the edges of the originally closed wound. Collagen material, unlike exposed dead bones, is a surface that may be prone to epithelialization, which gives the possibility of secondary wound closure as a result of granulation and epithelialization without re-exposing bone. The second argument is that the proven presence of numerous bacterial colonies in the area of sequestrants and gentamicin (released for a few days from the collagen carrier) shows the right spectrum, is directly applied, and acts at the site of inflammation [35]. Capar et al. conducted a study aimed at determining the efficacy of collagen sponges with doxycycline for the purposes of ONJ prevention after extraction. This study proves that the preliminary application of such sponges for extraction, socket, and to cover the wound can be an effective method of preventing osteonecrosis [59].

A questionable issue thus becomes the choice between the potential effect of topical antibacterial and antifungal lasers and conventional local drug therapy with consideration of antibiotics, which may be the main topic of many studies, both in vivo and in vitro.

The advantages and disadvantages of antibiotic use in MRONJ are summarized in Table 3.

### 2.6. Vitamin D

Vitamin D deficiency is a serious problem not only for ONJ patients, but also in all people living in areas with short periods of daylight days throughout the year. Its deficiency <20 ng/mL is found in 30% of young people and 80% of older people, of which 50–70% are middle-aged people, including postmenopausal women [38], which undoubtedly shows the scale and extent of the phenomenon and the fact that even small doses of supplementation of vitamin D should be a part of everyday life in the diet of middle-aged people, regardless of their state of health and disease status. Daily supplementation above 800 IU (international units) of vitamin D decreases twice the risk of advanced alveolar bone loss, which confirms the wide range of bone remodeling [60].

Vitamin D supplementation is still a subject of research; according to sources, the appropriate daily dose is considered to be in the range of 400 IU to 4000 IU [61]. Recent reports show that, in some cases, such supplementation is insufficient, and the appropriate recommended level is 40–80 ng/mL, which is difficult to achieve with, for example, 2000 IU [62,63]. According to research, a daily dose of 10,000 IU is completely safe and does not cause side effects. S. Ghanaati and J. Choukroun et al. developed a supplementation protocol in which people with vitamin D levels <40 ng/mL should supplement vitamin D at 10,000 IU daily. People with serum vitamin D concentration at the level of 40–80 ng/mL should take 5000 IU per day, while people with high levels >80 ng/mL should supplement vitamin D at a dose of 1000 IU [11]. Research in recent years has highlighted a number of benefits of vitamin D, not only in relation to the skeletal system, but also vitamins influencing the overall health and functioning of the body [64].

However, it is important to draw attention to its role in relation to MRONJ. In a study conducted by Badros et al. [65], as many as 40% of patients with multiple myeloma had vitamin D deficiency Vit. D < 36 nmol/L, and 35% of them showed a deficiency in the range of 36–75 nmol/L. The situation was in patients with breast cancer, where 30.2% had a vitamin D level below 50 nmol/L. The analysis of Akishike Hokugo et al. [66] carried out in animal models has shown that the combination of three factors, the reception bisphosphonate (zoledronic acid), tooth extraction, and vitamin D deficiency, determines the highest probability of developing drug-induced necrosis of jawbones. A group of laboratory rats deficient in vitamin D treated with zoledronic acid showed a 66.7% chance of developing ONJ, whereas in a group in which vitamin D was at the appropriate level with constant intravenous infusion, ZOL developed ONJ in 14.3% of cases. However, no cases of non- bisphosphonate control groups have been reported.

The important fact is that maintaining the concentration of vitamin D above 32 ng/mL (80 nmol/L) protects the body against the development of osteoporosis, cardiovascular diseases, and cancer. It is worth paying particular attention to the fact that vitamin D supplementation may protect against diseases that later require the use of bisphosphonates that can cause osteonecrosis of the jaw. This is a kind of primary prevention against the development of diseases, in the course of which the risk of ONJ increases dramatically [67,68]. Deficiency of vitamin D and calcium intake is one of the main causes of second male osteoporosis, as it has been proven that vitamin D levels below 30 ng/mL lead to an increased risk of developing osteoporosis and apparently interfere with osteoclastogenesis and bone remodeling. The state of increased risk of developing osteoporosis may indirectly be a factor increasing the risk of ONJ, and thus supplementation of vitamin D and maintaining a proper concentration may affect the development of osteoporosis and subsequently BRONJ [13]. The fact that an adequate level of vitamin D also has a preventive effect on such diseases as breast and prostate cancer is also significant [69]. Metastases of both tumors are often located in the bones, which are treated with bisphosphonates (intravenously).

One of the most important properties of vitamin D is a natural production of an antibiotics antimicrobial called natural peptide antibiotics. The special feature of these substances is the proangiogenic action, i.e., the exact opposite to the bisphosphonates, which, by inhibiting the production of the blood vessel network in the bones, contributes to necrosis [70] (Table 3).

Research shows, however, that not only vitamin D, but also the level of PTH, has a significant impact on the risk of osteonecrosis of the jaws. It has been shown that vitamin D deficiency induces disturbances in PTH and calcium levels [66]. One study evaluating the effect of PTH levels on the risk of ONJ found that people with elevated PTH levels, which is directly associated with vitamin D deficiency, were more likely to experience osteonecrosis of the jaws than patients with increased PTH [71]. A prospective clinical study showed a relationship between the level of immunoreactive parathyroid hormone (iPTH) and the risk of developing ONJ. Patients with significantly elevated iPTH levels (associated with vitamin D deficiency) before and during treatment with BP eventually suffered from ONJ. Conversely, in the case of patients with level iPTH before BP treatment, ONJs were not found in these patients [72]. Heim et al., in turn, conducted research on the level vitamin D and calcium among two different groups of patients taking antiresorptive drugs. The first group consisted of people with symptoms of ONJ in stage II. The second, constituting 18 patients, showed a lack of exposed bones in the oral cavity. Serum 25(OH)D concentrations were significantly higher in patients from group 2 (29.5 ng/mL vs. 20.49 ng/mL). Similar relationships were demonstrated with respect to calcium; the concentration in group 2 was 2.25 mmol/L, 0.11 SD versus 2.175 mmol/L, 0.16 SD. The results of this analysis show the dependence of ONJ on patients with low vitamin D levels and reduced absorption of calcium at a low of concentration vitamin D, which emphasizes the importance of proper supplementation of this vitamin in all persons receiving antiresorptive drugs [16].

## 3. Conclusions

Antiresorptive drugs are still underestimated by a large proportion of doctors. It should be stressed, however, that they play a significant role in the history of the disease and do not give way to drugs regulating cardiac or blood clotting, which could also cause dangerous complications. Bone artifacts are a topic that can occur in any dental practice, especially where surgical procedures predominate. Careful interview and preparation of the patient are some of the most important factors that could prevent this most complex situation. It is worth proposing therapy of increasingly frequent use of lasers biostimulation and Er:YAG, replacing the same conventional surgery. Simple biostimulation treatments, regardless of the stage of the disease, are able to improve healing and, in less advanced cases, lead to a complete cure. Using laser Er:YAG, in turn, numerous studies that have demonstrated efficacy equal to that of conventional surgery, while being softer for tissues. If the planned surgery does not have to be done overnight, it is worth getting the patient ready. For this purpose, it is recommended to check the level of vitamin D in the plasma and depending on the result correct supplementation. Vitamin D has a preventive effect and also positively affects the healing process of bone necrosis. It can be concluded that the implementation of supplementation in people with MRONJ should become obligatory because of the widespread deficiency of this vitamin, especially in our latitudes. Each case of MRONJ should be considered individually. Conventional antibiotic surgery is not always the only solution, and the development and reconstruction of the bones are influenced by so many factors that the treatment can begin by with oral supplementation vitamin D, which is the base for further, effective treatment.

## Figures and Tables

**Table 1 pharmaceutics-13-00354-t001:** Surgical treatment of medication-related osteonecrosis of the jaw (MRONJ). PRF, platelet-rich fibrin; PRP, platelet-rich plasma.

Surgical Recommended in Stage II or III Resistant to Non-Surgical Treatment; Usually Combined with Antibiotics		
	Combined with autogenous growth factors (like PRP, PRF)	Combined with hyperbaric therapy
Advantages	Biocompatibile autogenous GF (growth factors) improve healing, promote angiogenesis, and stimulate undifferentiated stem cells at the wound site differentiating towards osteoblasts	An effective and non- invasive support for the proper treatment
Weakness	Surgical treatment should be implemented in specific indications (as above) as an invasive method. May be problematic and severe for patients with systemic diseases (such as cardiovascular), in the elderly, and in poor general condition.	

**Table 2 pharmaceutics-13-00354-t002:** Non-surgical therapy with the use of lasers. LLLT, low-level laser therapy.

Therapy with the Use of Lasers			
	LLLT-Diode Lasers	Nd:YAG	Er:YAG
Advantages	Positive effect on the survival of osteoblast cells	Antibacterial and antifungal activity, reduction of inflammation and pain	Less thermal damage and faster healing compared with conventional surgery, and can be used for large and extensive surgical treatment
Weakness	Costs of treatment		

**Table 3 pharmaceutics-13-00354-t003:** Pharmaceutical support for MRONJ.

Pharmaceutical Therapy		
	Antibiotics	Vitamin D
Advantages	Eradication of a broad spectrum of bacteria	Production of an antibiotics antimicrobial called natural peptide antibiotics and vitamin D protects against development of osteoporosis, cardiovascular diseases, and cancer.
Weakness	Bacterial resistance to antibiotics, allergic reactions, intestinal flora disorders	Obtaining a toxic dose of 100 ng/mL is possible, but very difficult in practice

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
