# Peer review of "Schemes for Drug-Induced Treatment of Osteonecrosis of Jaws with Particular Emphasis on the Influence of Vitamin D on Therapeutic Effects"

_pharmaceutics, 2021, doi:10.3390/pharmaceutics13030354_

Round 1

Reviewer 1 Report

This manuscript described a very comprehensive review of therapeutic methods for treatment of osteonecrosis of jaws (MRONJ). The reviewer highly recommends the publication of this manuscript after minor revisions considering addressing the following questions:

  • The title should be modified. It is rare that a title used two sentences. Additionally, it seemed that vitamin D is not major discussion issue of this manuscript, though it is a new discussion point.
  • This manuscript does not have any Table or Figure to help understand the manuscript quickly. E.g. The authors may provide the table summarizing the different treatment methods for MRONJ as well as their respective strengths and weakness. That may make manuscript more attractive to potential readers.

Author Response

Answer to Reviewer #1:

Thank you for your consideration

The title was changed and summary tables completed

We hope that the response below will satisfy your expectations.

Yours sincerely,

Reviewer 2 Report

Dearest Authors,

nice work on an interesting topic, I would only suggest some minor improvements prior to publication:

1. increase the references also with respect to the use of antibiotics in other bone regeneration spaces

2. on vit D some nice works by Dr. J Choukroun should be referred

best greetings & stay safe

Author Response

Answer to Reviewer #2:

Thank you for your consideration and recommendation.

As suggested, the sentence was rewritten: Bibliography enriched with antibiotics and scaffold drug delivery systems. The content was enriched with the knowledge provided in the articles by Dr. J Choukroun

Reviewer 3 Report

The manuscript is well written and structured in the various points.

The authors performed wide literature review for non-surgical and surgical treatment of MRONJ, expecially in use of Vitamin D. 

References must be updated and extendeed, expecially for antibiotic paragraph.

It is necessary to correct some grammatical errors (for example in line 179, page 4 the point must be replaced with comma ).

Author Response

Answer to Reviewer #3:

Thank you for your consideration.

As suggested, the sentence was rewritten: the current knowledge and recommendations were verified and appropriate modifications to the manuscript were introduced.

Grammar: Existing grammar errors were corrected.

We hope that the response below will satisfy your expectations.

Yours sincerely,
